# Molecular Organization and Regulation of the Mammalian Synapse by the Post-Translational Modification SUMOylation

**DOI:** 10.3390/cells13050420

**Published:** 2024-02-28

**Authors:** Isabel Chato-Astrain, Marie Pronot, Thierry Coppola, Stéphane Martin

**Affiliations:** 1Université Côte d’Azur, CNRS, Inserm, IPMC, Sophia Antipolis, F-06560 Valbonne, France; ichatoastrain@ipmc.cnrs.fr (I.C.-A.); coppola@ipmc.cnrs.fr (T.C.); 2Centre for Discovery Brain Sciences, University of Edinburgh, Edinburgh EH8 9XD, UK; mpronot@exseed.ed.ac.uk

**Keywords:** post-translational modification, SUMO, SUMOylation, LLPS, biomolecular condensate, synapse

## Abstract

Neurotransmission occurs within highly specialized compartments forming the active synapse where the complex organization and dynamics of the interactions are tightly orchestrated both in time and space. Post-translational modifications (PTMs) are central to these spatiotemporal regulations to ensure an efficient synaptic transmission. SUMOylation is a dynamic PTM that modulates the interactions between proteins and consequently regulates the conformation, the distribution and the trafficking of the SUMO-target proteins. SUMOylation plays a crucial role in synapse formation and stabilization, as well as in the regulation of synaptic transmission and plasticity. In this review, we summarize the molecular consequences of this protein modification in the structural organization and function of the mammalian synapse. We also outline novel activity-dependent regulation and consequences of the SUMO process and explore how this protein modification can functionally participate in the compartmentalization of both pre- and post-synaptic sites.

## 1. Introduction

Neurons are polarized cells with one axon and several highly branched dendrites, which connect through synapses for rapid information transfer in the brain. Brain activity relies on the establishment of functional networks of interconnected neuronal cells. This process involves the accurate formation and maturation of synapses which are fundamental to establishing efficient neuronal wiring in the developing brain. The coordinated organization of complex protein networks on both sides of the synapse enables synaptic transmission and plasticity that supports cognitive processes. This structural organization encompasses many scaffolding and adaptor proteins and thus multiple protein–protein interactions to efficiently position the machinery for neurotransmitter release, post-synaptic neurotransmitter receptors and the associated signaling cascade components and their regulators.

Among the key regulatory mechanisms controlling the interactions and function of proteins within pre- and post-synaptic compartments are post-translational modifications (PTMs) including phosphorylation and ubiquitination. SUMOylation is another PTM playing important roles in the organization and function of the mammalian synapse. The SUMOylation process was first reported around 1996 [1,2] and for many years was exclusively associated with the nuclear compartment. Extranuclear SUMOylation is now well documented and occurs in all tissues, cells and organelles [3,4]. Although SUMOylation is also predominantly associated with the nucleus in neurons, it is also present in pre- and post-synaptic compartments [5,6,7,8,9].

SUMOylation is an enzymatic process that consists in the covalent but reversible isopeptidic binding of the Small Ubiquitin-like MOdifier (SUMO) to the lateral chain of specific lysine residues within target proteins. This modification alters the interacting properties of the SUMO-modified proteins through conformational changes, by preventing the binding of some proteins initially interacting with the non-SUMOylated form of the target protein, or by providing a binding platform for specific sets of new interacting proteins. The functional impact of SUMOylation is thus multiple and depends on the modified proteins. SUMOylation can affect the subcellular localization of the modified protein through its binding to novel protein partners or change its solubility, degradation or stability properties, but can also directly impact its intrinsic activity.

In this review, we discuss the most recent data on the SUMO-dependent regulatory mechanisms occurring at the synapse and consider the role of SUMOylation in maintaining the overall synaptic organization. We then highlight the activity-dependent function of specific SUMO-target proteins at both pre- and post-synaptic sites and outline the possibility that SUMOylation can assist liquid–liquid phase separation (LLPS) to efficiently achieve the compartmentalization of the mammalian synapse.

## 2. The SUMOylation/deSUMOylation Enzymatic Pathway at Synapses

SUMOylation occurs in all eukaryotic cells, and the consequences of this protein modification are diverse. Among all the cellular functions involving SUMOylation, we can cite cell proliferation, the maintenance of nuclear homeostasis and nucleocytoplasmic transport, the regulation of several cell signaling pathways including apoptosis and senescence, as well as the adaptative response to cellular stress [3,4].

The SUMO process as well as the associated enzymatic pathway leading to SUMOylation and deSUMOylation of target proteins in the brain have also been reviewed elsewhere [6,7,9,10,11]. Therefore, in this section, we first give a brief overview of the SUMOylation process and then review the data related to the synaptic distribution of the SUMO machinery and on the signaling pathways and mechanisms driving the regulation of the SUMOylation and deSUMOylation balance at the mammalian synapse.

### 2.1. SUMO Paralogs and the Associated Enzymatic SUMOylation/deSUMOylation Cascade

Small Ubiquitin-like Modifiers (SUMOs) are small proteins of ~100 amino acids (~11 kDa) sharing 18% sequence homology with Ubiquitin. Among the five known SUMO paralogs existing in mammals, SUMO1, SUMO2 and SUMO3, referred as SUMO2/3 due to their high homology (97% identity), are the only three isoforms expressed in the brain. Despite sharing less than 50% identity, the mature forms of SUMO1 and SUMO2/3 are conjugated to substrate proteins through a specific enzymatic pathway (Figure 1).

SUMOylation is highly dynamic and the equilibrium between SUMOylation and deSUMOylation is driven by a dedicated cascade of enzymatic reactions. First, the inactive SUMO precursors are matured by the hydrolase activity of certain SUMO proteases (SENP1,2,5) leading to the exposure of a C-terminal di-Glycine motif. Next, the mature form of SUMO (SUMO-GG) is activated by the heteromeric E1-activating complex formed by SAE1/SAE2 in an ATP-dependent manner, generating a thioester bond between the cysteine residue of SAE2 and SUMO-GG. The activated SUMO is then transferred to the active cysteine site of Ubc9, the only E2 SUMO-conjugating enzyme of the system. Ubc9 can now covalently promote the binding of the SUMO-GG to the lateral chain of the targeted lysine residue of the substrate protein, usually with the support of an E3 enzyme (Figure 1). SUMO E3s will help the SUMOylation process by enhancing the SUMO conjugation rate to the substrate protein.

Despite being covalently bound to SUMO, the SUMOylated target proteins can be efficiently deSUMOylated via the specific isopeptidase activity of any of the SENP enzymes (SENP1-3, SENP5-7). DeSUMOylation therefore leads to the release of free active SUMO moieties that are then available for additional rounds of SUMOylation (Figure 1).

### 2.2. Synaptic Localization of the SUMOylation/deSUMOylation Machinery

Although SUMOylation acts predominantly within the nuclear compartment or to allow the nucleocytoplasmic exchange of various target proteins, there is no doubt that SUMOylation can also occur at synapses regulating many important aspects of the pre- and post-synaptic function [6,7,8,9].

A group of recent publications reporting the partial localization of the components of the SUMOylation and deSUMOylation machinery at both pre- and post-synaptic sites in high-resolution microscopy [12,13,14] reinforced the previous findings that SUMOylation and deSUMOylation enzymes are indeed targeted to synapses [15,16,17,18,19,20,21]. Colnaghi and colleagues first used structured illumination microscopy to confirm that the immunoreactivities for SUMO1, SUMO2/3 and the SUMO-conjugating enzyme Ubc9 are preponderantly present in the nuclei of mouse hippocampal neurons as expected, but that a significant proportion of the labelling is also partially colocalized with pre- and post-synaptic markers [12]. They later reported that the deSUMOylation machinery, and in particular the deSUMOylation enzymes SENP1, SENP6 and SENP7, are also localized to some extent in the hippocampal synapses [13] in line with earlier reports. More recently, they further used super-resolution microscopy to show that the SUMO E3 ligases PIAS1 and PIAS3 are also expressed in pre- and post-synaptic compartments in mouse hippocampal neurons as for the other members of the SUMOylation machinery, but interestingly, this synaptic colocalization was much more limited in cortical neurons, suggesting distinct regulatory mechanisms driving the synaptic targeting and/or compartmentalization of the SUMO enzymes between different neuronal subtypes [14]. Importantly, the presence of the SUMO machinery at synapses does not prevent the possibility that some synaptic SUMOylated proteins could also result from their diffusion or targeting from extra-synaptic sites.

As the other enzyme of the SUMO system, SENP3 is mainly expressed in the nucleus but is also present in the synapses in the rat cerebrum and cerebellum [21]. SENP3 is involved in mitochondria fragmentation with a functional impact on the SUMOylation levels of the dynamin-related protein 1 Drp1 [22], but also by acting on the mitochondrial fission 1 protein (Fis1) deSUMOylation in mitochondrial autophagy [23]; both of these effects occur under stress conditions.

Interestingly, SENP enzymes present distinct SUMO preferences [24]. Among the SENP isoforms localized at synapses, SENP1 shows a slight preference for the removal of SUMO1 from SUMOylated substrates, whereas SENP3 preferentially deconjugates SUMO2/3 over SUMO1. Interestingly, SENP6 and SENP7 instead exert significant editing of the poly-SUMO2/3 chains. However, whether these different SENPs coexist within the same synapse and how they act on shaping the length, composition and/or complexity of the SUMO chains of specific synaptic substrates are still not known.

### 2.3. Synaptic Regulation of the SUMO Pathway

While a growing number of studies have demonstrated the functional consequences of specific SUMO substrates at synapses, few data are available in the literature regarding the mechanisms driving the targeting, anchoring and functional regulation of the SUMO enzymes at synapses.

Among the first evidence that the synaptic redistribution of the components of the SUMO system is regulated by changes in neuronal activity is the work achieved by the Henley group in 2009 [16]. They showed that the pharmacological stimulation of purified synaptosomal preparations by α-amino-3-hydroxy-5-methyl-4-isoxazolepropionic acid (AMPA), a specific agonist of the ionotropic glutamate AMPA receptors (AMPAR) prior to a subsynaptic fractionation, leads to a significant increase in the association of the SUMO-conjugating enzyme Ubc9 in the presynaptic active zone. This redistribution of Ubc9 to the presynaptic region following AMPA receptor stimulation was accompanied by an increase in the total level of SUMOylated proteins in the presynaptic fraction [16].

The same group later showed that the mRNA levels for both SUMO1 and Ubc9 are increased in dendrites following a chemical protocol to trigger long-term potentiation (LTP). This pharmacological treatment also led to an increased immunoreactivity for both SUMO1 and Ubc9 at post-synaptic sites, suggesting an activity-dependent local translation mechanism for the SUMO pathway [25].

Consistent with an efficient regulation of the SUMOylation machinery at synapses, we reported the rapid presynaptic accumulation of the AoS1/SAE1 SUMO-activating enzyme and SUMO-conjugating enzyme Ubc9 upon a short KCl depolarization in rat hippocampal neurons [18]. Concomitantly, the presynaptic level of deconjugating enzyme SENP1, but not SENP6, was transiently decreased. Post-synaptic AoS1 and Ubc9 protein levels were both rapidly decreased upon depolarization resulting in a transient decrease in protein SUMOylation in synaptoneurosomes. This synaptic drop in synaptic SUMO substrates occurs without affecting the overall SUMOylation levels, thus revealing an activity-dependent regulation of the SUMO system directly within the synaptic compartments [18].

To better understand how the SUMO system is regulated at synapses, our group started using restricted photobleaching and photoconversion experiments on individual dendritic spine from cultured hippocampal neurons expressing a GFP-tagged form of Ubc9 [19]. We measured the diffusion properties of the SUMO-conjugating enzyme in and out of spine upon variation of synaptic activity with pharmacological drugs. We showed that the diffusion of Ubc9 into spines is regulated through an mGlu5R-dependent signaling pathway involving the mobilization of calcium ions and PKC activation. This activation promotes a transient phospho-dependent synaptic trapping of Ubc9 and consequently an increase in its residency time at the post-synapse, promoting synaptic SUMOylation events. Interestingly, this synaptic diffusional trapping observed in mGlu5R-activated conditions is lost when using the K65A-Ubc9 mutant, which, contrary to the WT form of Ubc9, is unable to discriminate between phosphorylated and native substrates [19]. This work thus revealed that the dynamic diffusion of Ubc9 is subject to activity-dependent regulatory processes and provides the first mechanism leading to changes in synaptic SUMOylation levels during mGlu5R activation (Figure 2).

Additionally, live imaging studies from our laboratory revealed a more complex regulation of the SUMO system at the mammalian synapse. Indeed, we demonstrated that the deSUMOylation enzyme SENP1 targeting and anchoring properties at post-synaptic sites are also dynamically driven by the activation of type I mGluRs (Figure 2; [26,27]). In real-time imaging approaches on individual spines expressing GFP-tagged SENP1 coupled to subcellular fractionation, we found that the activation of mGlu5R leads to a time-dependent decrease in the exit rate of SENP1 from dendritic spines, resulting in the post-synaptic accumulation of the deSUMOylation enzyme [26]. More recently, we added another level of regulation by showing that mGlu1 receptor activation works in opposition to mGlu5R. We demonstrated that mGlu1R acts as a brake to the mGlu5R-dependent deSUMOylation process at the post-synapse [27]. Preventing the activation of mGlu1R during type 1 mGluR stimulation evokes a much faster and larger post-synaptic accumulation of SENP1 (Figure 2). Altogether, these findings reveal that type I mGlu1/5 receptors are central to dynamically maintaining the homeostasis of SUMOylation at the mammalian synapse but also indicate that the relative amount of mGlu1 and mGlu5 receptors at individual synapses will drive the diversity of post-synaptic SUMOylation responses. However, what functional impact this bidirectional regulation may have on specific post-synaptic substrates and how it may affect synaptic transmission and plasticity are still open questions.

As highlighted above, few studies so far have investigated the balance between SUMOylation/deSUMOylation at synapses. Indeed, the molecular and signaling pathways by which this spatiotemporal equilibrium is achieved and activity-dependently regulated at synapses are still in their infancy. However, they deserve much more attention to better understand the physiological and pathophysiological consequences of this post-translational modification in the brain [6,8,9].

## 3. SUMOylation in Neurite Growth, Synapse Formation, Elimination and Maturation

Brain activity is involved in the formation and remodeling of synapses formed during brain development. The establishment of functional neuronal circuits in the developing brain relies on neural stem cell differentiation and upon the formation, elimination and maturation of synapses. The role of SUMOylation in neural stem cell differentiation and survival has been reviewed very recently [28]. In the following section, we go over the data involving SUMO target proteins important for establishing neuronal polarity and dendritic branching complexity, as well as for synapse formation and stabilization (Table 1).

### 3.1. SUMOylation of Transcription Factors in Neurite Growth and Branching

Neurite growth and branching requires the spatiotemporal activation of several transcription factors, including, among others, members of the Myocyte Enhancer Factor 2 MEF2, the Forkhead box P2 (FOXP2) or ZBTB20, a member of the Broad complex, Tramtrack, and Bric-à-brac/poxvirus and zinc finger (BTB/POZ) family of transcriptional repressors. The following subsections highlight the functional roles of these transcription factors’ SUMOylation on neurite growth and branching.

#### 3.1.1. MEF2 SUMOylation

The transcription factor family of Myocyte-specific enhancer factor 2 (MEF2) is central to brain development and particularly abundant in the cerebellar cortex [29]. MEF2s play critical roles in cell differentiation, dendritic morphogenesis, synapse formation, pruning and synaptic plasticity. Mutations of these genes are associated with various pathological conditions, including epilepsies and autism spectrum disorders, among others [30]. The function of MEF2 is regulated by several PTMs, including acetylation, phosphorylation and SUMOylation.

SUMO1-ylation of MEF2A at the K403 residue inhibits its transcription activity and thus promotes the differentiation of dendritic protrusions [31]. The engineering of SENP2 knockout embryos and the use of in vitro SUMO assays in SHSY5Y cells showed that the deSUMOylase SENP2, but not SENP1, regulates MEF2A deSUMOylation in response to activity-dependent stimuli [32]. Co-expression of SENP2 and MEF2A leads to the deSUMOylation of MEF2A and to an increase in its transcriptional activity [32]. These data highlight that the SUMOylation/deSUMOylation equilibrium is important for regulating the transcriptional activity of MEF2A. Interestingly, MEF2A can also be acetylated on the same K403 residue, resulting in an increase in MEF2A transcriptional activity and consequently to the inhibition of dendritic differentiation and synapse disassembly [31]. Shalizi and colleagues demonstrated that the switch between SUMOylation and acetylation of MEF2A involves calcium signaling and the Serine 408 dephosphorylation close to the SUMO/acetylation acceptor site. This calcium signal leads to the calcineurin-induced dephosphorylation of the S408 residue on MEF2A followed by the deSUMOylation and acetylation of the K403 residue, thereby regulating the transcription of MEF2A target genes that orchestrate the post-synaptic complexity [31]. In addition, MEF2A SUMOylation has also been reported to participate in presynaptic differentiation in rat brains [33]. The expression of the SUMOylated transcriptional repressor form of MEF2A drives the suppression of orphan presynaptic sites in vivo via a direct repression of the gene encoding for the presynaptic protein Synaptotagmin [33].

#### 3.1.2. FOXP2 SUMOylation

The transcription factor Forkhead box P2 (FOXP2) is SUMOylated at K674 (K673 in mouse) [34,35,36]. Mutations within the *FOXP2* gene lead to brain developmental abnormalities with reduced gray matter associated with speech and language deficits. In an elegant study, Usui and collaborators characterized the SUMOylation of FOXP2 in the developing mouse cerebellum and showed that it modifies transcriptional regulation by FOXP2 [36]. In utero electroporation of specific shRNAs to *FOXP2* in Purkinje cells led to a significant reduction in dendritic outgrowth and arborization. This reduction of dendritic outgrowth and arborization in Purkinje cells that was rescued by the WT expression of FOXP2 but not by its SUMO-deficient mutant KR revealed the functional impact of FOXP2 SUMOylation in neuronal differentiation through neurite/dendritic outgrowth and arborization. Interestingly, they also showed that a reduction in FOXP2 expression in the cerebellum significantly alters the neonatal righting reflex and negative geotaxis and that both phenotypes were rescued by the WT FOXP2 complementation, but not by the SUMO-deficient FOXP2 mutant expression, supporting the role of FOXP2 SUMOylation in cerebellar motor functions [36].

Interestingly, FOXP1 SUMOylation is also SUMOylated on the K670 residue and rapidly deSUMOylated upon NMDA receptor stimulation [37]. The dendritic length is severely reduced upon FOXP1 knockdown in cultured mouse cortical neurons and cannot be rescued upon expression of the non-SUMOylatable FOXP1-K670R mutant. These data confirm that the SUMOylation level of FOXP1 is central to the development of dendritic arborization in the developing brain.

#### 3.1.3. MeCP2 SUMOylation

Methyl-CpG-binding protein 2 (MeCP2) acts as a transcriptional repressor, and mutations within the human *MECP2* gene on the X chromosome cause a severe neurodevelopmental disease in females called Rett syndrome. SUMOylation of the MeCP2 protein at the K223 was initially reported [38], and this modification is essential for its transcriptional repression activity in mouse primary cortical neurons. Using in utero electroporation of specific MeCP2 RNAi in the developing rat hippocampus reduced the density of excitatory synapses. This neurodevelopmental defect was fully rescued upon the co-expression of the WT form of MeCP2 in the rat hippocampus, but not when the non-SUMOylatable K223R form of MeCP2 was expressed [38].

More recently, the K412 residue was identified as an additional SUMOylation site on MeCP2 [39]. The authors elegantly demonstrated that the phosphorylation of MeCP2 at S421 and T308 facilitates its SUMOylation on K412. They also showed that the SUMOylation of MeCP2 is upregulated via the activation of NMDA, IGF-1 and CRF signaling pathways, ultimately resulting in an increased *bdnf* mRNA expression via a CREB-dependent mechanism. To assess the functional consequences of MeCP2 SUMOylation in vivo, the authors used lentiviral particles to express either the WT MeCP2, its non-SUMOylatable K412R mutant or a SUMO fusion form of MeCP2 in the brain of *Mecp2* KO mice. Strikingly, while the WT and MeCP2-SUMO forms were able to rescue the deficits in social interaction and fear memory measured in *Mecp2* KO mice, re-expression of the non-SUMOylated K412R-MeCP2 mutant was not [39]. Altogether, the above data strongly support a key functional activity-dependent role for MeCP2 SUMOylation in the synaptic development of the central nervous system.

#### 3.1.4. ZBTB20 SUMOylation

ZBTB20 is a zinc finger protein and a transcriptional repressor [40]. ZBTB20 is involved in neurite growth and branching in the developing brain. Defects in the function of ZBTB20 is linked to a wide range of neurodevelopmental disorders, including intellectual disability and autism [41,42].

Ripamonti and colleagues reported the SUMOylation of the ZBTB20 protein at the K330 and K371 residues [43]. They showed that the lack of ZBTB20 SUMOylation has no effect on its nuclear localization or dimerization. Its SUMOylation instead modifies the interacting pool of transcriptional co-regulators, leading to a change in ZBTB20-target gene expression. Deletion of *Zbtb20* in hippocampal neurons results in a decrease in the complexity of the neurite branching in immature neurons but also in more mature cells, revealing a functional role for this translational repressor in the control of neurite growth and ramification. Overexpression of the WT form of ZBTB20 in *Zbtb20*-lacking neurons is able to rescue neurite branching defects, while re-expression of its non-SUMOylatable mutant cannot, further suggesting a functional role for the SUMOylation of the ZBTB20 protein in neurite growth and branching [43]. However, it would have been of interest to also analyze how the overexpression of the non-SUMOylated form of ZBTB20 in WT neurons would impact the length and ramification of neurites by acting as a dominant negative over the native form of ZBTB20 to reinforce the finding that ZBTB20 SUMOylation is important for neurite growth and arborization.

**Table 1 cells-13-00420-t001:** Summary table of the impact of SUMOylation of specific transcription factors on neurite growth and branching.

Transcription Factor	SUMOylation Site	Effect of SUMOylation	References
MEF2A	K403	Inhibits transcription activity, promotes dendritic claw differentiation	[31,32,33]
FOXP2	K670, K673/674	Transcriptional regulation and control of dendritic arborization	[36]
MeCP2	K223, K412	Repression of MeCP2 transcriptional activity, impact on spine density	[38,39]
ZBTB20	K330, K371	Affects the transcriptional activity of ZBTB20 and acts on neuritogenesis	[43]

### 3.2. Extranuclear SUMOylation in Synapse Formation and Maturation

Extranuclear SUMOylation also acts significantly on axonal growth and the formation, density and maturation of dendritic spines, which is essential to establish a functional neuronal network. Hereafter is summarized some of the work achieved to better understand the role of SUMOylation in synapse formation and stabilization (Table 2).

#### 3.2.1. CASK SUMOylation

CASK (Ca2+/Calmodulin-dependent serine protein kinase) is a member of the membrane-associated guanylate kinase (MAGUK) family and plays an important role in the formation of dendritic spines through its interaction with the actin cytoskeleton via the adhesion protein 4.1 [44]. Knockdown of the endogenous CASK expression in hippocampal neurons results in a reduction in the density, length and width of dendritic spines via its interaction with the protein 4.1 [45]. They also showed that SUMOylation of CASK at K679, a site very close to the protein 4.1 binding site, impairs its interaction with the protein 4.1, which in turn reduces the association of CASK with the actin cytoskeleton. Interestingly, overexpression of a fusion CASK-SUMO1 protein to mimic CASK SUMOylation leads to a sharp decrease in spine density, length and width, suggesting that the SUMOylation state of CASK is important for the stabilization of mature dendritic spines via its impact on adhesion molecules and actin cytoskeleton [45].

#### 3.2.2. Local Protein Synthesis and Dendritic SUMOylation

The formation and maintenance of synapses require local protein synthesis and thus the transport of specific mRNAs along dendrites. The correct targeting of these mRNAs and their activity-dependent local translation regulation are also crucial for synaptic function. In the following section, we describe how SUMOylation participates in the functional regulation of RNA-binding proteins (RBPs) essential to synaptic transmission and plasticity.
CPEB3 SUMOylation

The RNA-binding CPEB3 belongs to a family of four proteins in vertebrates and is involved in the regulation of local translation in the brain, and consequently in synaptic plasticity and memory formation (for a recent review on the role of CPEB protein in learning and memory, see [46]). The Kandel lab first reported the SUMO2-ylated CPEB3 in hippocampal neurons both in vitro and in vivo [47]. They showed that the SUMOylation of CPEB3 regulates its oligomerization. In a basal state, CPEB3 is SUMOylated and acts as a local translational repressor. CPEB3 is rapidly deSUMOylated upon neuronal activation, leading to its aggregation and the translation of target mRNAs, including the *sumo2* mRNA [47]. In an additional study, they showed that CPEB3 is localized to P-bodies that are discrete membraneless cytoplasmic condensates enriched in translationally arrested mRNAs [48]. They further demonstrated that upon chemically induced long-term potentiation, CPEB3 promotes phase separation from P-bodies and is then translocated to polysomes, allowing the translation of target mRNAs. When SUMOylation is pharmacologically inhibited, CPEB3 colocalizes less with P-bodies in line with the importance of CPEB3 SUMOylation in the distribution of the protein to these membraneless domains and consequently in the ability of CPEB3 to repress mRNA translation locally [48]. The above findings thus suggest a central role for the SUMOylation state of CPEB3 to regulate local mRNA translation via the control of its distribution, oligomerization and activity during synaptic plasticity events.

Additional computational studies were recently reported to better understand the dynamics of SUMO2 binding to CPEB3 to regulate translational control in dendritic spines [49,50]. In these publications, the authors first propose a bioinformatic model for the interaction between CPEB3 and actin filaments (F-actin) in which the CPEB3/F-actin interaction might be potentially regulated by CPEB3 SUMOylation [49]. Then, they proposed an additional computational model showing the complex structural interaction between the RNA-binding domain of CPEB3 and SUMO2, suggesting that the allosteric effect of this modification may impact the affinity of RNA-binding to CPEB3 [50].
FMRP SUMOylation

The RNA-binding protein FMRP participates in the formation and organization of membraneless RNA-containing structures in dendrites and axons to transport and locally regulate the translation of several mRNAs essential for synaptic formation and plasticity in a mGlu5 receptor-dependent manner [51,52,53,54]. Loss of FMRP expression or function leads to the most common inherited cause of intellectual disability and autism called Fragile X Syndrome (FXS; [53,54]). At the cellular level, the absence of FMRP leads to a pathological hyperabundance of immature spines in FXS patients and mouse models of the disease [55,56]. The consequences of these synaptic defects are synaptic transmission and plasticity alterations leading to socio-cognitive impairments. FMRP is the target of several PTMs, including phosphorylation, ubiquitination and methylation, that are essential to tune the function of FMRP in neurons (for a comprehensive review, see [57]). More recently, FMRP has been identified as a SUMOylation target in the mammalian brain revealing an additional level of complexity in the regulation of the FMRP function [58]. The SUMOylation of FMRP is triggered by the activation of mGlu5 receptors in line with the known regulation of the SUMOylation/deSUMOylation balance via group 1 mGluRs [19,26,27]. SUMOylation of FMRP can occur on K88, K130 and K614, but the functional activity of FMRP SUMOylation lies within the two proximal lysine residues. A short activation of mGluR5 (<5 min) triggers the SUMOylation of FMRP in dendrites, leading to its dissociation from mRNA granules at the bases of dendritic spines and allowing the release and local translation of target mRNAs essential to the elimination and maturation of selected immature spines [58]. Consequently, expression of a non-SUMOylable N-terminal K88,130R FMRP mutant in mouse neurons exerts a dominant negative effect and a phenotype similar to that reported for FXS patients, characterized by increased density of immature dendritic protrusions [58]. Interestingly, state-of-the-art spatiotemporal proteomics approaches have identified the composition and organization of stress granules as well as the mechanisms regulating granule dissociation. In this work, the authors notably demonstrated that the dual mutation of the K88 and K130 SUMO sites on FMRP also impairs stress granule formation and dissociation, confirming the importance of these residues and their SUMO modification for the function of FMRP [59].

More recently, Yang et al. identified a de novo heterozygous gene-truncating mutation of the deSUMOylation SENP1 gene in patients with ASD [60]. In line with the previous published data [58], they nicely demonstrated that the decrease in SENP1 expression in *Senp1* haploinsufficient mice leads to an increase in the level of FMRP SUMOylation and to its subsequent degradation. The viral re-introduction of the WT form of FMRP, but not its non-SUMOylatable K88,130R mutant, in the brain of these mice nicely rescued the social deficits and repetitive behaviors observed in the heterozygous *Senp1*^+/−^ mice. Similar results were obtained upon the viral re-expression of SENP1 in the mutants to compensate the low level of expression due to haploinsufficiency [60], further highlighting the preponderant role of FMRP SUMOylation in the developing brain.

### 3.3. SUMOylation and Microtubules (MTs)

Microtubules (MTs) are abundant in neurons occupying axons and dendrites to provide a structure allowing them to acquire or maintain their specialized morphologies [61]. MTs are also present in spines, and knockdown of the MT-plus-end-binding protein EB3 significantly reduces spine formation, confirming the role of MTs in the regulation of dendritic spine development [62]. Molecularly, MTs are polymers of tubulin heterodimers consisting of α-tubulin and β-tubulin subunits interacting with high affinity. The polymerization/depolymerization of MTs are governed by a mechanism called dynamic instability [63], which involves several post-translational modifications of tubulin as well as interactions with various MT-associated proteins (for a comprehensive review, see [61]). Recently, several MT-associated proteins have been identified as SUMO targets.

The Katanin p60 ATPase-containing subunit A1 (KATNA1) is a microtubule-cleaving enzyme which has been reported to be SUMO2-ylated at the K330 residue [64]. At the molecular level, KATNA1 hydrolyses ATP to release energy to sever MTs [65]. Overexpression of KATNA1 in hippocampal neurons leads to an increased neurite length and promotes branching [66], while knockdown of KATNA1 inhibits axonal growth [67]. Overexpression of the SUMO-deficient K330R KATNA1 mutant in cultured hippocampal neurons decreases the total neurite length as well as the axonal and dendritic length [64]. This indicates that the SUMO-modification at K330 in KATNA1 enhances the severing activity of the enzyme, promoting neurite outgrowth by cleaving acetylated MT.

Spastin is another MT-severing protein shown to be SUMO-modified at K427 [68]. The non-SUMOylatable K427R impacts the Spastin conformation and in turn abolishes the ability of Spastin to cleave MTs, leading to their stability. Interestingly, overexpression of the non-SUMOylated K427R Spastin mutant in cultured hippocampal neurons led to an increased number of mushroom-like spines, suggesting that the balance between the SUMOylation and deSUMOylation of Spastin plays a role in the maturation of dendritic spines [68]. Altogether, these data highlight a differential role for the SUMOylation KATNA1 and Spastin on MT dynamics.

KATNA1 and Spastin exert differential actions in axonal growth [69]. While SUMOylation of KATNA1 promotes axonal growth, nothing has been reported regarding the role of Spastin SUMOylation in axonal formation and/or branching. It would thus be of interest to evaluate the effect of Spastin SUMOylation on axonal growth to gain further insights into the role of SUMOylation in MT-driven axonal formation.

Interestingly, α-tubulin was identified as a SUMO substrate in mass spectrometry screening in 2005 [70]. More recently, Feng and collaborators reported the preferential SUMOylation of the soluble form of α-tubulin by SUMO1 at K96, K166 and K304, close to the loops involved in contacts between protofilaments [71]. The SUMOylation of α-tubulin is reversed upon SENP1 activity. Mutation of the three SUMO sites into Arginine residues fully abolishes the SUMOylation of α-tubulin. Using MT assembly–disassembly assays, they assessed the role of α-tubulin SUMOylation on MT dynamics and showed that SUMOylation of α-tubulin reduces MT assembly and promotes MT catastrophe, consequently leading to the increased instability and reduced length of the MT. They also looked at the role of SUMOylation of α-tubulin in the neuron-like cell line N2A, and observed a decrease in neurite length upon the expression of the non-SUMOylated form of α-tubulin. However, the impact of α-tubulin SUMOylation in primary cultured neurons is still unknown. More recently, the SUMOylation of α-tubulin was also identified using mass spectrometry from SUMO2/3-enriched synaptic preparations [72]. Due to the essential role of MT dynamics for neurite growth, extension and branching, and the association of disrupted MT stability with neurological disorders such as microcephaly and neurodegenerative diseases [73]), investigating how SUMOylation of α-tubulin regulates neurite growth in the mammalian brain will certainly provide new insights into the physiological and pathophysiological mechanisms behind the complex regulation of MT dynamics in the brain.

## 4. SUMOylation, Biomolecular Condensates and Compartmentalization of the Synapse

The consequences of specific SUMOylated proteins at the synapse have been reviewed extensively in recent years [6,7,8,9]. In this section, we instead discuss the concept of *en masse* SUMOylation at the synapse and then highlight the current findings as well as the potential roles for some SUMO substrates in partitioning the synaptic compartments through the formation of biomolecular condensates and how synaptic activity may regulate phase transition.

### 4.1. En Masse SUMOylation at Synapses?

The physiology and physiopathology associated with the equilibrium between the SUMOylation and deSUMOylation of nuclear proteins has been extensively reviewed [74,75], but several proteins are also SUMO-modified in extranuclear compartments in neurons [6,7,8,9]. However, given the low proportion of SUMOylation outside of the nucleus and the high SUMO protease activities [76], identification of SUMOylated proteins in other subcellular compartments and especially at synapses remains particularly challenging. Recently, we combined the use of subcellular fractionation experiments on brains obtained from 14-day post-natal rat pups followed by denaturing immunoprecipitation with specific anti-SUMO2/3 antibodies and mass spectrometry analysis, allowing the identification of 803 endogenous synaptic SUMO2/3-modified proteins [72].

This unique dataset represents about 18% of the synaptic proteome and includes adhesion molecules, vesicle trafficking and cytoskeleton-associated proteins, scaffolding proteins and neurotransmitter receptors and transporters. These potential SUMO2/3-ylated targeted proteins have also been functionally associated with synaptic processes like synapse formation, synaptic vesicle cycling and neurotransmission and synaptic plasticity.

The interesting concept of ‘*en masse* SUMOylation’, also referred to as ‘Protein-Group SUMOylation’ emerged from quantitative mass spectrometry (MS) experiments primarily aiming to investigate DNA repair [77]. These data revealed that SUMOylation most frequently targets entire groups of proteins that are physically interacting or in an immediate proximity, rather than individual proteins. Given the number of SUMOylation target proteins identified by MS in the restricted space of the synapse [72], it is tempting to expand this concept to the synaptic compartments where specific clusters of proteins are likely to undergo simultaneous activity-dependent rounds of SUMOylation, allowing additional SUMO-interacting motif (SIM)–SUMO interactions and the formation of functional sub-synaptic biomolecular condensates (Figure 3).

### 4.2. SUMOylation and LLPS

A growing number of studies now report potential roles for protein SUMOylation in liquid–liquid phase separation (LLPS) and in the formation of biomolecular condensates in response to various cellular stresses (see [11,78] for recent reviews on the roles of SUMOylation in phase separation and biomolecular condensates).

LLPS, phase transitions and formation of biomolecular condensates are interconnected events with essential roles in the organization of specific subcellular domains and are thus detrimental processes to cell function. Despite extensive investigation in recent years, little is known about how the formation and dissociation of these biomolecular condensates are regulated, especially within the synapse. Interestingly, there is increasing evidence for the pivotal role of SUMOylation in LLPS-based formation/dissociation of these biomolecular condensates [11,78]. The best-described role for SUMOylation in LLPS is linked to the nucleus, where SUMOylation regulates the formation of promyelocytic leukemia protein (PML) bodies [79]. Formation of these biomolecular SUMO-containing condensates is partly based on the SUMO from a substrate protein and its non-covalent interaction via the SIM of another protein, bridging many proteins and their bound molecules together into membraneless condensates. SIM domains are usually composed of three exposed hydrophobic residues close to negatively-charged residues allowing the non-covalent binding of SUMO and are thus present in many different proteins [80,81]. SIM thus represents a prevalent and well-characterized motif involved in the recognition of SUMO and mediating non-covalent protein–protein contacts in the context of SUMOylation by increasing the allosteric interactions between SUMO-modified and non-SUMOylated proteins [82].

### 4.3. SUMOylation and Compartmentalization of Pre- and Post-Synaptic Sites

Presynaptic compartmentalization is essential to ensure the activity-dependent and calcium-sensitive control of neurotransmitter release. The mechanism of exocytosis has been studied for many years, which was detrimental to defining the precise role of the proteins involved and to understanding the sequence of molecular events occurring at each step of the process [83]. The release of neurotransmitters in the synaptic cleft is the direct consequence of axonal plasma membrane depolarization, which leads to the opening of presynaptic calcium channels. The increase in presynaptic calcium concentration triggers neurotransmitter exocytosis via complex and rapid molecular reorganization events. The precise control of ion input and output is therefore essential to this process. In the past few years, numerous studies have shown that several types of ion channels are SUMOylated [84]. The overall presynaptic SUMOylation appears as an important modulator of synaptic membrane potential and excitability which is central to coordinating neurotransmitter exocytosis [9]. Several voltage-sensitive (Kv) potassium channels required for membrane repolarization are SUMOylated [9,84]. In a simplistic way, these presynaptic SUMOylation events lead to a decrease in the overall K^+^ conductance. Sodium channels Nav1.2, which trigger action potentials in neurons, are also SUMOylated and their SUMO modification regulates the velocity of backpropagating action potential in neurons and thus directly impacts synaptic transmission [85]. In addition, SUMOylation of the Collapsin Response Mediator Protein 2 (CRMP2) directly impacts the influx of calcium ions into neurons [86] through its interaction with the N-type Voltage-Gated Calcium Channel (VGCC) Cav2.2, but also via the regulation of Nav1.7 sodium channels [87] by holding them at the plasma membrane. As discussed above, FMRP is activity-dependently SUMOylated [58] and importantly can also interact directly with the C-terminal domain of Cav2.2 channels to modulate its cell-surface expression and activity [88]. Interestingly, the functional regulation of Cav2.2 channels by SUMOylation at the K394 residue was also reported [89]. By modulating the level of protein SUMOylation in superior cervical ganglion (SCG) neurons, the authors measured an increase in the ratio of paired excitatory post-synaptic potentials [89], which is consistent with an increase in the neurotransmitter release probability, further involving VGCC SUMOylation in the control of presynaptic function.

Recent evidence has emerged to support the concept that resting state, exocytosis of neurotransmitter-containing vesicles and endocytosis occur in separate spaces within the presynaptic bouton, and are organized into distinct protein clusters via LLPS, including synapsin condensates for clustering synaptic vesicles (SV) [90], RIM/RIM-binding protein networks to shape the active zone [91] and sub-synaptic protein clusters forming the endocytic sites [92]. All these coordinated presynaptic events and the associated protein clusters are summarized in Figure 4. A number of presynaptic proteins central to the regulation of exocytosis have been reported [7,8,9].

The first example in the literature is the SUMOylation of synapsin-1a [95]. Synapsins ensure that SVs are maintained in a reserved accessible pool which is essential for delivering SVs to the active zone of the synapse. The authors demonstrated that the SUMOylation of synapsin-1a at K687 is required to maintain the size of the reserve pool of SVs and that the SUMO-modified form of synapsin-1a interacts more efficiently with exocytosis vesicles [95]. Synapsin-rich condensates recruit additional SVs or soluble proteins including Synaptophysin, VGLUT1 and α-synuclein [101,102], which are known as SUMOylated or potential SUMO targets [72,99]. More recently, Hoffmann and collaborators used two-color single-molecule tracking and super-resolution microscopy to confirm that synapsin-1 and SVs can form condensates to control the accumulation of SVs at presynaptic terminals [103]. Then, they showed that this process is highly dynamic since SVs maintain their motility in synaptic boutons despite being restricted in subdomains by synapsins. This indicates that the formation of synapsin-1 condensates is important to regulate both the clustering and the motility of SVs. However, nothing is known about the effect of synapsin SUMOylation on the dynamics of its interactions at presynaptic sites.

At the center of the exocytosis machinery are VGCCs, the site of calcium entry, and synaptotagmin-1, which is the calcium sensor triggering SV exocytosis. The exocytosis machinery on the presynaptic side is highly regulated by post-translational modifications. Thus, the activity of several kinases, including the well-described PKA, PKC and CaMK, is crucial for modulating neurotransmitter release [104].

RIM1α is a Rab3a effector protein, capable of multiple interactions with key synaptic proteins, including VGCC, syntaxin-1a, SNAP25 and synaptotagmin-1, to shape the active zone [105]. These interactions are regulated by calcium ions, inositol phosphates or phosphorylation. Interestingly, the SUMOylation of RIM1α at K502 concentrates ion channels in the active zone [93], whereas its non-SUMOylated form is instead involved in vesicle priming.

Synaptotagmin-1 is a calcium sensor essential for fast SV exocytosis reported to be SUMO1-ylated [94]. To date, the functional consequences of this SUMOylation remain elusive but may be linked to the clustering of molecules around the neurotransmitter releasing site. Indeed, it was recently shown that synaptotagmin-1 couples the presynaptic exocytic SV fusion to the endocytic retrieval of SV by promoting the local formation of a kinase/lipid signaling complex [106], but whether the SUMOylation of synaptotagmin-1 at presynaptic sites could be involved in this functional activity-dependent clustering or its interaction with RIM1α is still not known.

**Table 2 cells-13-00420-t002:** Summary of SUMO substrates and their associated functions highlighted in the review.

Target Protein	SUMOylation Site	Effect of SUMOylation	References
CASK	K679	Prevents interaction with protein 4.1 and the association of CASK with the actin cytoskeleton; control of dendritic spine density	[45]
CPEB3	K50, K294	Regulates its oligomerization and acts as a local translational repressor	[47,48,49]
FMRP	K88, K130, K614	Triggers the mGlu5R-dependent dissociation of FMRP-SUMO from mRNA granules, leading to local translation of mRNAs essential to spine maturation and elimination	[58]
KATNA1	K330	Enhances the activity that cleaves acetylated microtubules, leading to neurite outgrowth	[64]
Spastin	K427	Abolishes the ability to cleave MTs, thus impacting their stability and consequently spine maturation	[68]
α-Tubulin	K96, K166, K304	Reduces microtubule assembly affecting their length and subsequently neurite growth in neuronal cell lines	[71,72]
CRMP2	K374	Interaction with Voltage-Gated Calcium Channels and anchorage of Nav1.7 to the plasma membrane	[86]
Cav2.2	K394	Modulates neurotransmitter release by the activation of the presynaptic Ca^2+^ channels	[89]
RIM1α	K502	Concentrates ion channels in the active zone; required for vesicle exocytosis	[93]
Synaptotagmin-1	?	Increases Synaptotagmin-1 SUMOylation in transgenic mice specifically overexpressing SUMO1 in neurons	[94]
Synapsin-1a	K687	Maintains the synaptic vesicles in a reserved accessible pool ready to be activity-dependently delivered	[95]
Syntaxin-1A	K252, K253, K256	Reduces the interaction with SNAP-25 and VAMP-2 SNARE proteins	[96]
mGluR7	K889	DeSUMOylation of mGlu7R leads to its internalization	[98]
GluK2	K886	Promotes the activity-dependent endocytosis of GluK2-containing Kainate receptors	[15,107,108]

Many synaptic proteins including SUMOylated proteins also possess SIM domains that non-covalently bind SUMO moieties, and thus may facilitate complex assemblies by trapping other SUMO-modified proteins in their vicinity, consequently leading to clustering of specific SIM/SUMO-proteins in sub-synaptic domains (Figure 3). SUMOylation might therefore have additional functions, by organizing subdomains of the synapse, in shaping the protein content of presynaptic condensates to achieve a proper SV cycle and neurotransmitter release (Figure 4).

Another key protein of the SNARE protein complex essential for SV exocytosis and endocytosis is syntaxin-1. The syntaxin-1A is SUMOylated within its C-terminal region at K252, 253 and K256, close to the transmembrane domain [96]. The interaction of the SNARE proteins SNAP-25 and VAMP-2, but not Munc-18, with syntaxin-1A, is significantly reduced by its SUMOylation. Using real-time presynaptic imaging experiments, the authors also showed that the SUMOylation of syntaxin-1A instead regulates the rate of SV endocytosis without impacting the exocytosis process [96].

The case of the type III metabotropic glutamate receptor mGluR7 is particularly interesting [98]. The amount of glutamate released at the synapse is modulated by a feedback control loop involving mGluR7. The phosphorylation of mGluR7 at S862 facilitates its SUMOylation at K889 [98]. In addition, cell incubation with a mGluR7 agonist leads to the deSUMOylation of mGluR7 and to its internalization. Similarly, overexpression of the deSUMOylating enzyme SENP1 increases the endocytic rate of the receptor [98]. These data suggest that maintaining mGluR7 at the membrane is the best way to preserve negative feedback on the glutamate release to prevent any potential depletion of the pool of neurotransmitter vesicles ready for exocytosis.

Strikingly, 12% of SUMO2/3-modified proteins recently identified in a synaptic SUMO mass spectrometry screening [72] are associated with the localization, transport or recycling of synaptic vesicles, extending the role of SUMO2/3-ylation beyond just regulating the formation of condensates at the presynaptic site to include regulation at each step of the SV cycle. Indeed, proteins modulating SV docking and priming like Rab3c and Rab5 and the exocytosis events like VGCCs or proteins associated with vesicular membranes including VGLUT1, synaptophysin and SV2B, as well as proteins implicated in SV recycling such as V-ATPase and dynamin, have also been identified as potential SUMO2/3 targets [72].

This widespread SUMOylation of SV proteins raises the hypothesis that *en masse* SUMOylation of vesicle-associated proteins may represent an efficient way to dynamically change the protein composition of sub-synaptic complexes to rapidly maintain the vesicular pool and synchronize neurotransmitter release and SV recycling within the presynaptic compartment (Figure 4). However, the precise roles for SUMOylation on the localization and function of these newly identified presynaptic SUMO targets remain to be addressed.

The process of protein cluster/phase separation is not restricted to the compartmentalization of the presynaptic site and can be extended to the post-synaptic compartment. Indeed, recent exciting studies have reported that the assembly and activity-dependent modulation of the post-synaptic density (PSD) composition involves LPPS [109,110,111]. The PSD is composed of hundreds of different proteins, including neurotransmitter receptors, scaffold proteins and many signaling molecules, with these dynamically interconnected assemblies creating a defined synaptic subdomain. It has been shown that four major post-synaptic proteins, PSD-95, SAPAP, Shank and Homer, form the PSD condensate [109]. Formation of condensed molecular assemblies has also been suggested for AMPAR synaptic clustering and transmission [110]. The binding of PSD95 to transmembrane AMPAR regulatory proteins (TARPs), which are essential AMPAR auxiliary subunits, helps to regulate the number and density of surface-expressed AMPARs at the PSD and consequently strengthen the synapse [112]. In addition, several enzymes such as SynGAP interact with PSD95, increasing their concentration within the constrained space of the PSD [109]). Upon synaptic stimulation, SynGAP is dispersed from the PSD condensates, highlighting the dynamic nature of PSD assemblies [113]. The mechanisms by which the assembly and disassembly of PSD condensates are regulated to allow changes in post-synaptic surface-expressed AMPARs are still unclear. However, 7% of the identified SUMO target proteins at synapses (55 out of 803) are directly linked to the PSD [72], including the major PSD components PSD95, Homer, SAPAP and SynGAP1 (Figure 5).

Given the recent identification of a large set of SUMO targets at the synapse and the molecular consequences of SUMOylation in synaptic transmission and plasticity, further investigations to address the involvement of this protein modification in the activity-dependent assembly and disassembly of presynaptic and PSD condensates are urgently needed. However, assessing the real-time SUMOylation states of multiple proteins in or out of the synaptic clusters and evaluating their physiological impact are still not quite feasible with the molecular and imaging tools currently available.

## Figures and Tables

**Figure 1 cells-13-00420-f001:**
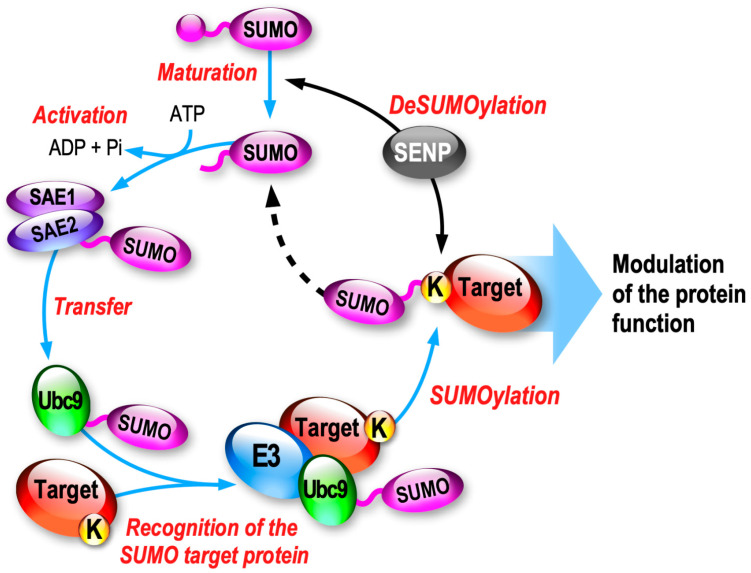
Schematic representation of the enzymatic SUMOylation/deSUMOylation cycle. The different phases of the enzymatic cascade leading to the SUMOylation of the target protein are highlighted in red. SUMO, Small Ubiquitin-like MOdifier; SAE, SUMO-activating enzyme; Ubc9, SUMO-conjugating enzyme; SENP, Sentrin protease.

**Figure 2 cells-13-00420-f002:**
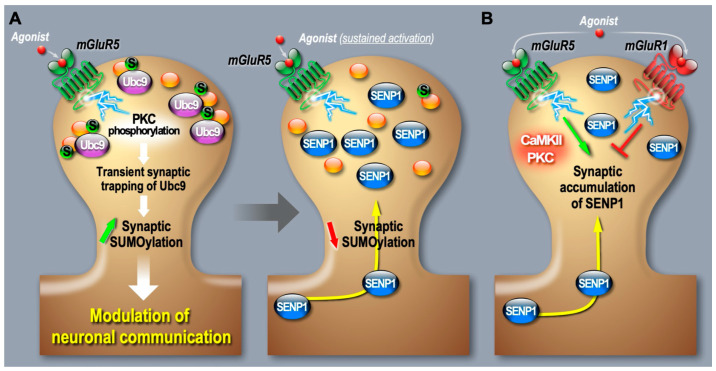
Post-synaptic regulation of the SUMO/deSUMOylation cascade at synapses by type I mGlu1/5R. (**A**) A short activation of mGlu5 receptors triggers a Protein Kinase C (PKC) phosphorylation-dependent trapping of the sole SUMO-conjugating enzyme Ubc9 at post-synaptic sites, leading to an increased SUMOylation and to the modulation of neuronal excitability [19]. The sustained activation of mGlu5R then leads to a decrease in the exit rate of the deSUMOylation enzyme SENP1 from dendritic spines, resulting in the post-synaptic accumulation of SENP1 to bring SUMOylation back to initial levels [26]. (**B**) The post-synaptic accumulation of SENP1 upon stimulation of mGlu5R involves the downstream activation of PKC or Ca^2+^/Calmodulin-dependent protein kinase II (CaMKII). Blocking mGlu1R enhances the post-synaptic accumulation of SENP1, indicating that the SUMOylation/deSUMOylation balance is bidirectionally regulated by type I mGluRs to control the levels of synaptic SUMOylation [27].

**Figure 3 cells-13-00420-f003:**
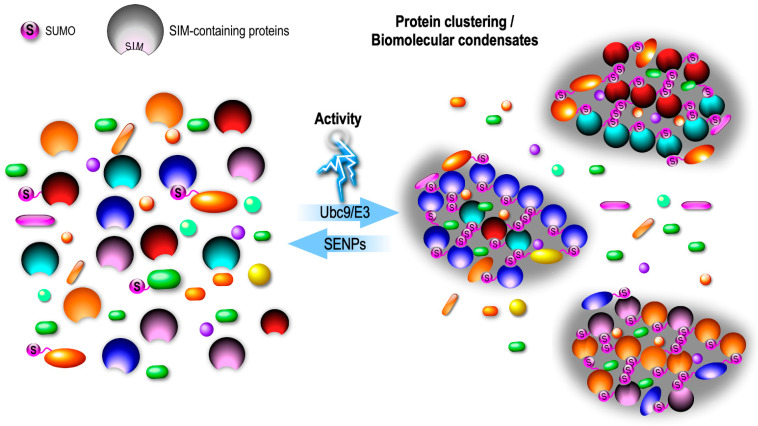
Potential mechanism driving the activity-dependent protein clustering into distinct subdomains via SUMO-interacting motif (SIM)/SUMO-protein-dependent interactions.

**Figure 4 cells-13-00420-f004:**
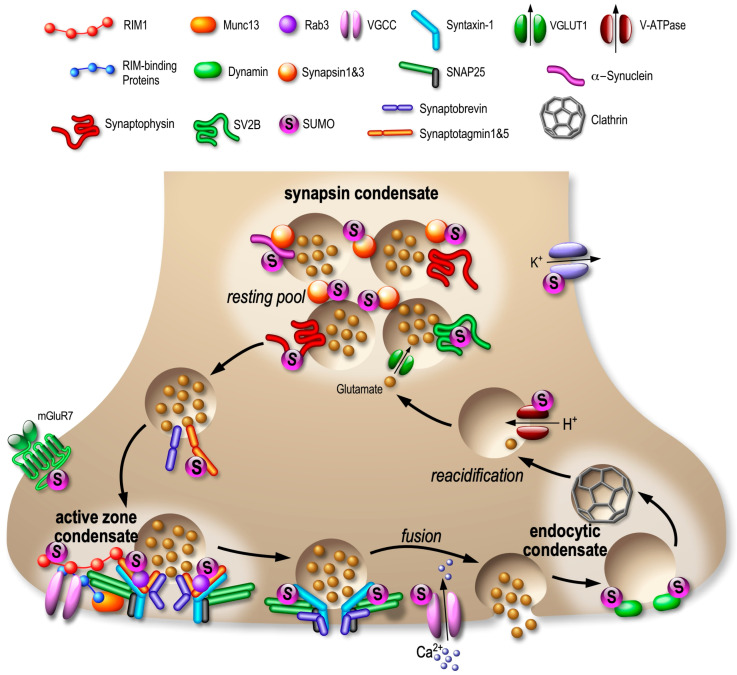
Landscape of SUMOylated proteins in the compartmentalization of presynaptic sites. Consistent with the emerging presynaptic functions of SUMOylation, several presynaptic proteins are SUMO substrates: RIM1α [93], Synaptotagmin-1 [94], Synapsin-1 [95], Syntaxin-1 [96], mGluR7 [97,98] and α-synuclein [99,100]. In addition, Rab3, SV2B, Synaptophysin, VGCCs, VGLUT1, V-ATPase and Dynamin have been identified as potential SUMO targets in [72].

**Figure 5 cells-13-00420-f005:**
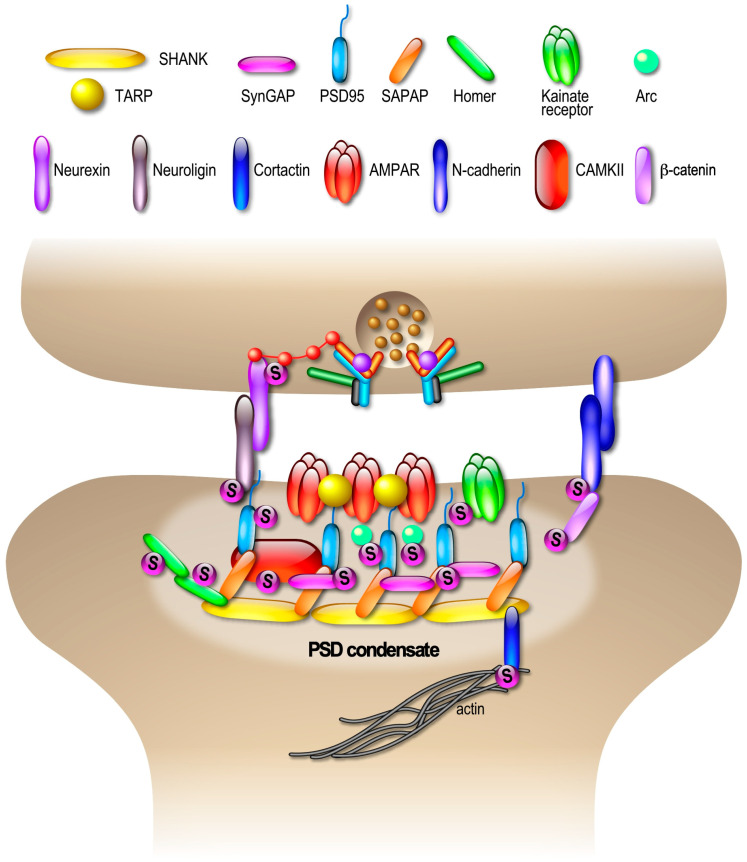
Landscape of SUMOylated proteins in the compartmentalization of post-synaptic sites. Consistent with the emerging post-synaptic functions of SUMOylation, several post-synaptic proteins are known SUMO substrates, including Kainate receptors [15,107,108] and Arc [114,115]. In addition, SynGAP, SAPAP3, PSD95, Homer1, CaMKII and Adhesion molecules like Neuroligin, N-Cadherin and β-catenin have been identified as potential SUMO targets in [72].

## Data Availability

Not applicable.

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
