# Peer review of "Molecular Organization and Regulation of the Mammalian Synapse by the Post-Translational Modification SUMOylation"

_cells, 2024, doi:10.3390/cells13050420_

Round 1
Reviewer 1 Report
Comments and Suggestions for Authors
The manuscript gives us a comprehensive summary about the regulation of sumoylation in structural organization and function of the mammalian synapse.This paper is well constructed and suitable to be published when issues solved and questions in this manuscript are addressed:
1. Provide a clear transition from the general discussion of synapse structure and function to the specific focus on SUMOylation in introduction.
2. Consider defining "SUMOylation" and briefly explaining its relevance in the introduction itself. This can help readers unfamiliar with the term grasp its significance early on..
3. Consider including a table to summarize sumoylated proteins and their functions to help the readers better understand it.
4. In phase separation part, better use subtitle to list the sumoylated proteins and give a summary of the regulation.
Comments on the Quality of English LanguageThe sentences are well constructed.
Author Response
We thank the reviewer for his/her insightful comments. Here are enclosed our point-by-point answers to the referee’s request.
- Provide a clear transition from the general discussion of synapse structure and function to the specific focus on SUMOylation in introduction.
We have changed the text to address the point made (lines 36-39).
- Consider defining "SUMOylation" and briefly explaining its relevance in the introduction itself. This can help readers unfamiliar with the term grasp its significance early on.
We have included an additional paragraph to address this remark in the revised version of the manuscript (lines 45-54).
- Consider including a table to summarize sumoylated proteins and their functions to help the readers better understand it.
We thank the reviewer for raising this point. We choose not to add an extended table here since a comprehensive and complete review (Henley et al, 2021; Ref 9) already summarized the synaptic SUMOylated proteins and their associated functions. In addition, we have also reported recently the identification of 803 proteins using mass spectrometry experiments on synaptic preparations (Pronot et al, 2022; ref 72 in this manuscript). In this last publication, an extended excel table (Supp Table 3) is already provided with all the potential SUMO-modified proteins identified at the rat synapse. We thus feel that providing another table describing all these proteins would not be useful to the reader as they already exist in the literature.
We rather decided to include a first table regrouping SUMOylated transcription factors highlighted in the review that have a direct impact on neuritogenesis (Table 1, page 8) and another one, with the remaining SUMO substrates discussed in this manuscript (Table 2, page 11).
- In phase separation part, better use subtitle to list the sumoylated proteins and give a summary of the regulation.
We thank the referee for this suggestion. However, we do think that it would be better not to make a list of all the proteins impacted by SUMOylation and potentially involved in synaptic clustering. Indeed, we want here to put forward the exciting hypothesis that SUMOylation at synapses could participate in phase separation and/or protein clustering and to the functional compartmentalization of pre- and post-synaptic sites.
Reviewer 2 Report
Comments and Suggestions for Authors
The review Article “Molecular organization and regulation of the mammalian synapse by the post-translational modification SUMOylation” by Isabel Chato-Astrain et al shed light on the role of SUMOylation on synapsis organization, regulation of various activities, and interaction of multiple proteins. This specific PTM ensures the proper functioning of synapsis and regulates their transmission and plasticity. The paper is well-organized and written. I have the following comments that will help to improve the review further.
1. Line 66-76: this paragraph tried to explain Figure 1 but somehow fell short in synchronizing the figure with the text. It could be due to Figure 1 is not well labeled. I will recommend labeling the different domains in Figure 1 for Clarity.
2. Figure 2: Although the Authors attempt to explain the mechanism of activation and regulation of SUMOylation, but it is hard to understand the figure. For example, it is not clear what does red or green arrow means. They depicted the involvement of CaMKII in the figure but has not been discussed in the text. It will be good to have a summary in figure legend.
3. Including a table summarizing the SUMOylation of various transcription factors will be helpful.
4. Figure 3: The author proposed the formation of various clusters in a SIM-SUMO-dependent manner, but they also showed the presence of a cluster that is free from any SIM-containing protein (blue shape). Please include any known mechanism that involves this cluster formation but is free of SIM-containing protein. Also, the corresponding mechanism is not well explained in the text.
Author Response
We thank this reviewer for the supportive comments. S/He states that “The paper is well-organized and written.” Here are enclosed the answers to the referee’s comments:
- Line 66-76: this paragraph tried to explain Figure 1 but somehow fell short in synchronizing the figure with the text. It could be due to Figure 1 is not well labeled. I will recommend labeling the different domains in Figure 1 for Clarity.
The different phases of the SUMO cascade have been highlighted in red in the figure and a short legend has been included for a better understanding of the figure (lines 98-101).
- Figure 2: Although the Authors attempt to explain the mechanism of activation and regulation of SUMOylation, but it is hard to understand the figure. For example, it is not clear what does red or green arrow means. They depicted the involvement of CaMKII in the figure but has not been discussed in the text. It will be good to have a summary in figure legend.
We thank the reviewer for this remark. We have now included a legend to help the reader (lines 201-211).
- Including a table summarizing the SUMOylation of various transcription factors will be helpful.
We have now included a Table showing the impact of SUMOylation of the specific transcription factors discussed in the review to help the readers (Table 1, page 8).
- Figure 3: The author proposed the formation of various clusters in a SIM-SUMO-dependent manner, but they also showed the presence of a cluster that is free from any SIM-containing protein (blue shape). Please include any known mechanism that involves this cluster formation but is free of SIM-containing protein. Also, the corresponding mechanism is not well explained in the text.
We thank the reviewer for bringing this mistake to our knowledge. We changed the SIM-missing proteins in the new figure 3. Indeed, the figure was initially designed to draw the reader’s attention about the exciting possibility that SIM-SUMO dependent interactions could lead to protein clustering. We also add few words to better explain this in the legend of the figure 3.
Reviewer 3 Report
Comments and Suggestions for Authors
The manuscript covers a highly relevant topic of the regulation of cellular processes by PTMs. SUMOylation is one of such modification that may cause substantial structural rearrangement of protein complexes and thus regulate various aspects of cell physiology. Authors describe SUMOylation and deSUMOylation role in modulating protein-protein interaction in brain synapses.
The manuscript is interesting to read, contains summary on various aspects of SUMOylation of synaptic proteins and I believe it is useful for the specialists in the field. The paper can be accepted without any major changes. However, some general information on SUMOylation role, outside of the brain specifically, should be added in the beginning of Section 2 or as a subparagraph, without just mentioning that it “have been extensively reviewed”. Plus, some summaries can be added in each subsection, so they do not seem just as reciting of dozens of studies, but more as their formal analysis. Whereas the parts on specific proteins as well as experimental designs and techniques (“LLPS” should not clearly be a keyword as well as “synapse” and “PTM” for being too general) used for their investigation can be shortened. This is especially important for sections 2.3, 3.2 and 4.3. The ones in the beginning and end of these parts are not clear really and do not actually help “to better understand the role of SUMOylation in synapse formation and stabilization”. A table showing which proteins (with their functions) from sections 2, 3 and 4 are upregulated and which are downregulated by SUMOylation, will help a lot. And if possible, please indicate, whether the formation of SUMOylated protein cluster actually induce or prevent the effective neurotransmission (if depends, what factors may determine the result).
P.S. The meaning of SIM abbreviation is revealed after it is first introduced in the text.
Comments on the Quality of English LanguageMinor typos ("reported to SUMO1-ylated") should be checked and corrected.
Author Response
We thank the reviewer for his/her strong support. S/He states “The manuscript is interesting to read, contains summary on various aspects of SUMOylation of synaptic proteins and I believe it is useful for the specialists in the field. The paper can be accepted without any major changes.” S/He also proposes a few suggestions to improve the manuscript. The referee's comments are in red.
However, some general information on SUMOylation role, outside of the brain specifically, should be added in the beginning of Section 2 or as a subparagraph, without just mentioning that it “have been extensively reviewed”.
We have now included some text regarding some of the broad cellular roles of SUMOylation (lines 62-66).
Plus, some summaries can be added in each subsection, so they do not seem just as reciting of dozens of studies, but more as their formal analysis. Whereas the parts on specific proteins as well as experimental designs and techniques (“LLPS” should not clearly be a keyword as well as “synapse” and “PTM” for being too general) used for their investigation can be shortened. This is especially important for sections 2.3, 3.2 and 4.3. The ones in the beginning and end of these parts are not clear really and do not actually help “to better understand the role of SUMOylation in synapse formation and stabilization”.
We thank the reviewer for this remark and we have included an additional table to summarize the roles of the SUMO-modified proteins discussed in the review including those from sections 2 and 3.
We are not sure to understand the argument that the keywords are too general since we have chosen to focus our review on the synaptic roles of SUMOylation, a post-translational modification, and its potential involvement in LLPS at synapses. We believe that, in addition to SUMO and SUMOylation, the keywords “LLPS”, “synapse” and “PTM” are summarizing the concepts we wanted to cover in this review.
We are also not sure to understand the second point raised here for sections 2.3, 3.2 and 4.3 since only the sub-part 3.2 focused on synapse formation and stabilization. However, the reviewer is right that the last part of section 3.2 was misplaced and did not help the reader to understand the role of SUMOylation in synapse formation and stabilization. We have therefore dissociated this last part from section 3.2 and make it as a separate section highlighting the role of SUMOylation on microtubules. This has been included in the revised manuscript as the Section 3.3 (line 438).
A table showing which proteins (with their functions) from sections 2, 3 and 4 are upregulated and which are downregulated by SUMOylation, will help a lot. And if possible, please indicate, whether the formation of SUMOylated protein cluster actually induce or prevent the effective neurotransmission (if depends, what factors may determine the result).
We have now included two tables in the revised manuscript. A first table showing SUMOylated transcription factors from the review that have a direct impact on neurite development (Table 1, page 8) and a second one, regrouping the remaining SUMO substrates discussed in the other sections of the current manuscript (Table 2, page 11). We could not answer the last point as there is no study yet showing the impact of SUMO-induced protein clusters on synaptic transmission.
P.S. The meaning of SIM abbreviation is revealed after it is first introduced in the text.
This has been changed in the revised manuscript
Minor typos ("reported to SUMO1-ylated") should be checked and corrected.
This has been fixed in the revised manuscript